# Work Engagement among Prison Officers. The Role of Individual and Organizational Factors in the Polish and Indonesian Penitentiary Systems

**DOI:** 10.3390/ijerph17218206

**Published:** 2020-11-06

**Authors:** Andrzej Piotrowski, Ewa Sygit-Kowalkowska, Imaduddin Hamzah

**Affiliations:** 1Institute of Psychology, University of Gdańsk, Jana Bażyńskiego 4 St., 80-309 Gdańsk, Poland; 2Department of Psychology, Kazimierz Wielki University, Leopolda Staffa 1 St., 85-867 Bydgoszcz, Poland; esygit@ukw.edu.pl; 3Community Guidance, Politeknik Ilmu Pemasyarakatan, Jl. Raya Gandul Cinere4, 16514 Depok, Indonesia; imaduddin@poltekip.ac.id

**Keywords:** prison officers, work engagement, workplace well-being, coping strategies, health

## Abstract

The literature on work engagement among prison officers (POs) remains rather scarce, and there are no analyses on the factors determining this phenomenon. The current study aimed to examine the relationships between work engagement, subjective well-being, coping strategies, and organizational factors utilizing the Utrecht Work Engagement Scale (UWES-9), the Coping Orientation to Problems Experienced (COPE), and Cantril’s Ladder of Health Scale (CLHS), and involving 312 POs from Poland and 467 POs from Indonesia. Results showed a statistically significant relationship between active coping and work engagement in both groups. Subjective well-being was moderately related to work engagement among Polish POs. Mean work engagement and subjective well-being scores were higher among Indonesian POs. The analyses showed a significant indirect effect of subjective well-being for the relationship between penitentiary unit type, active coping, as well as avoidant behaviors and work engagement in the Polish group. Closed prison officers more often declared higher subjective well-being. Work engagement is a complex psychological phenomenon. There exists a justified need for the analyses to consider personal determinants (e.g., coping strategies) as well as organizational factors related to the POs’ work environment. The literature presents a broad picture of the benefits of studying this phenomenon.

## 1. Introduction

The specific qualities of professional work places specific demands on employees and can put them at risk of various physiological and mental strains, namely, stress. The work of a prison officer (PO), in an institution that is separate from the rest of society, is described in the literature as potentially heavily stressful and having a significant, negative impact on wellbeing [1,2,3,4,5,6,7,8]. On average, POs are characterized by poorer health and higher suicide rates compared to police officers and the general working population [9,10,11]. Workplace stressors identified thus far in the prison context are, among others, authoritarian relationships with one’s superiors, a low level of participation in organizational decision-making, contact with aggressive inmates, and physical and verbal assaults from the inmates [12,13,14,15]. Symptoms of depression are also more often observed among POs who are in direct contact with inmates compared to those who are not (e.g., supply workers) [16]. Researchers of health in the context of prison work point to the fact that POs’ work absences and low productivity are related to workplace stress and the general characteristics of prison work [17,18].

Interest in the processes of coping with stress has increased since the 1980s [19]. The starting point for most of the research in this domain was the attempt to analyze stress and coping proposed by Lazarus and Folkman. They posited that stress is comprised of three processes: primary appraisal, secondary appraisal, and coping. The most common classification of coping strategies distinguishes problem-focused, emotion-focused, and avoidance strategies.

When exposed to difficult workplace situations, individuals engage in various coping strategies. These are defined as individual efforts directed at managing demands which exceed capabilities or cause strain [20]. It is noted that individual ways of coping can significantly impact the emergence of stress and emotional burnout (exhaustion) caused by work, as well as influence the self-assessment of one’s achievements [21,22]. Emotion-focused and avoidant (passive) coping strategies are positively related to the intensity of perceived stress [23]. As a result, so-called maladaptive coping strategies are also distinguished [19]. They provide short-term relief and distance the individual from stress [24]. Examples include procrastination, distraction, and substance abuse [25]. In turn, problem-solving, seeking instrumental support, or identifying priorities in action have an adaptive character [26].

Thus far, research has established that an appropriate reaction to difficult situations in penitentiary units can have a positive impact on the psychological and social functioning of their staff [27,28]. Social support in the penitentiary environment is a protective factor against stress and occupational burnout [12]. Coping strategies such as positive reinterpretation were also shown to reduce stress symptoms in a sample of Malaysian POs. Among people exhibiting high stress levels, common coping strategies include focusing on and venting emotions, denial, behavioral disengagement, and substance use [29]. The work–family life balance can serve a buffering role [30]. Research results show that this balance is difficult to maintain in the PO’s line of work. However, little remains known about whether coping strategies can influence the development of positive phenomena, such as work engagement, among POs. In turn, this knowledge has significant practical implications.

Work engagement is a state opposite to occupational burnout. It is characterized by a belief in one’s ability to meet the demands of one’s job. In W. Schaufeli and A. Bakker’s concept, work engagement is a positive sense of fulfillment, a feeling of enthusiasm, and contentment [31,32,33]. Employee engagement has become one of the most popular concepts in management in recent years. The concept of work engagement is closely related to positive psychology, health analysis in organizations [34], as well as the classic theories of work satisfaction which aim to increase work effectiveness through identifying the benefits of work, in contrast to the traditional focus on studying pathology [35]. Unlike work satisfaction or organizational loyalty, the notion of engagement in the context of work and effectiveness first appeared in research on consulting businesses, and only afterwards in the sphere of workplace and organizational psychology and management sciences [36]. The practical aspects of this approach are related to a very low-cost increase in effectiveness, causing many companies to implement solutions to assess and facilitate employee engagement. From the perspective of the employee, work engagement is related to their personal growth, while from the perspective of the organization, it improves its functioning. Minimizing the risk of occupational burnout is a particularly important aspect of engagement.

Work engagement is related to initiative, going beyond one’s job role, higher job performance, commitment to the organization [36], innovation [37], higher goal orientation [38], productivity [39], organizational citizenship behavior [40], creativity [41], and sharing of knowledge with coworkers [42]. Other analyses of work engagement also showed it has a negative impact on absenteeism, deviant behavior, and turnover intention [43]. Thus, the attitude or behaviors of work engagement are desirable within the prison service. The available research results point towards other emotional states among POs in their work environment. As the data shows, they experience anger and frustration, as well as fear and guilt. They also report that they have become more cynical and hardened as a result of their work [44]. Research on work engagement among POs remains scarce.

There exists a range of challenges resulting from the characteristics of penitentiary systems in given countries. As reflected in Polish studies, the scope of current difficulties involves a high proportion of repeated offences or a high proportion of inmates per one PO [45]. In turn, in the Indonesian penitentiary system, problems include an occupancy level of 174.2% (30.06.2020 for 2018) and a high number of inmates, exceeding 256 thousand [46]. The majority of Indonesian research in English concerns inmate problems (health) and general problems (overcrowding, terrorism and deradicalization, drug-related crimes) rather than PO problems. In contrast to the Indonesian penitentiary system, the occupancy level in Poland is 86.3% (31.5.2020—not including prisoners temporarily outside prisons). The prison population in Poland reached over 72 thousand in 2019 [47]. Simultaneously, this number is high compared to other EU countries (e.g., Germany or France) [48]. The proportion of inmates per 100,000 citizens is 183 in Poland and 85 in Indonesia [49]. Since 2010, the number of inmates in Poland has decreased by 6 thousand, while in Indonesia, it has increased by over 120 thousand.

Becoming a prison officer in the Polish Prison Service requires passing psychological assessment and a physical fitness test. Formal conditions must also be met, such as the lack of a criminal record or having at least secondary education. The training system of the prison officers in the Polish Prison Service comprises the following courses: introductory training, basic training in training centers, specialist training, and on-site training. Higher-ranking personnel—sub-officers and officers—are trained in the Central Prison Service Training Center in Kalisz. Stress prevention training, interpersonal training, and holiday retreats are organized in order to combat the negative consequences of work, in addition to promoting sport and healthy dietary habits [50]. Additionally, medical examinations assessing fitness for continued service are mandatory. Work in the Polish Prison Service is not considered prestigious.

In Indonesia, prison personnel are recruited by way of selection, which includes assessing documents according to formal criteria, civic knowledge, intelligence, and personality assessments, as well as health and physical endurance tests. These are carried out by medical, army, and/or police personnel. Psychological interviews are additionally conducted. After passing the selection, future prison personnel members undergo basic training for civil service workers. Prison officers undergo advanced training in prison security and order. Prison employees are not included in any special physical and psychological care programs. Maintaining physical and psychological health is their personal responsibility. However, twice per year, the General Directorate of Penitentiary Institutions organizes a sports competition among the personnel. Mental health programs are usually carried out as part of spiritual counseling. Prison Service work in Indonesia is a prestigious profession.

Penitentiary personnel is a part of an organizational culture, which is “a model of values, norms, beliefs, attitudes, and assumptions which do not have to be formalized, but which shape behavior and ways of realizing tasks” [51]. Due to a highly hierarchical structure and an authoritarian style of management, the organizational culture in the Prison Service is an additional source of stress for the personnel [52]. Research points to a significant work burden, health problems, and occupational burnout in prison personnel which result from the organizational culture [50,53]. Prison personnel highlights external factors affecting motivation and job satisfaction, the most important ones being adequate pay and other material benefits. They are meant to compensate the difficult work conditions, which rarely bring satisfaction in themselves [54]. Research shows that a poorly fit motivational system causes low work engagement and high turnover [55].

In the context of negative phenomena, it seems pertinent to analyze the underpinnings of the positive phenomenon of work engagement emerging despite a stressful work environment.

The aim of the current study was to identify the determining factors of work engagement in a group of Polish and Indonesian POs. The analyses included:-The influence of factors defined as individual: age and occupational inheritance (representing a subsequent generation working in the uniformed services), strategies of coping with job stress, and subjective well-being,-The influence of organizational factors, namely, penitentiary unit type (open, closed, semi-open), single- or multi-shift work.

The study sought to answer the following research questions:What are the work engagement and subjective well-being levels, as well as the coping strategies of POs? Are there differences between the Polish and Indonesian groups?Do individual and organizational factors impact work engagement levels? Are there differences between the Polish and Indonesian groups?Which of the analyzed variables are dignificant predictors of work engagement levels among POs in each of the two countries?

## 2. Materials and Methods

### Ethics Approval

On 28 January 2020, the study received the approval of the Ethics Committee of the Faculty of Psychology of the Kazimierz Wielki University in Bydgoszcz, Poland, as well as of the administrations of the penitentiary units taking part in the study. The study was carried out after obtaining the consent of the individual penitentiary units’ administrations. Participation in the study was anonymous and voluntary.

Due to the specificity of the functioning of the Prison Service, a random selection of participants is very difficult. Obtaining consent for carrying out research is often related to identifying specific penitentiary facilities where the research can take place. For this reason, the study was carried out on a convenience sample. Nevertheless, care has been taken to carry out the research in penitentiary facilities and custody suites which are representative in terms of size, system of functioning, type (open, half-open, closed), and inmate population.

The study utilized the following measures:(1)Coping Orientation to Problems Experienced (COPE) by Ch. C. Carver, M. F. Scheier, and J. K. Weintraub [56]. The questionnaire assesses 15 strategies of coping in stressful situations. It is based on self-assessment and it comprises 60 statements, with answers given on a 4-point Likert-type scale from 1 (*I usually don’t do this at all*) to 4 (*I usually do this a lot*). Individual coping strategies can be aggregated into more general strategies [57,58,59]. In the current study, a method of aggregation was chosen that would simultaneously fit the data from the Polish and the Indonesian group. The strategies were divided into three groups:(a)Active coping, comprising the strategies of active coping, planning, suppression of competing activities, positive reinterpretation and growth, and restraint,(b)Avoidant coping, comprising the following six strategies: behavioral disengagement, humor, substance use, acceptance, and denial.(c)Seeking social support and emotion-focused coping, comprising the following four strategies: use of instrumental social suport, use of emotional social suport, religious coping, and focus on and venting of emotions.(2)Cantril’s Ladder. A measure of psychological well-being—Ladder of Health Scale by H. Cantril (1965) [60]. It graphically represents numbers from 0 to 10. In the current study, the number 0 denoted the poorest possible health, while the numer 10 denoted the best possible health. The participants estimate their health in the current moment and make an X sign on a corresponding place on the ladder.(3)Utrecht Work Engagement Scale (UWES-9) by Wilmar Schaufeli and Arnold Bakker (2003), measuring general work engagement and its three consitutent parts [61]. Its three subscales are: vigor, dedication, and absorbtion. The current study utilized a 9-item version of this measure (with three items for each subscale). The questionnaire is comprised of items referring to the respondent’s job and their well-being as employees. Answers are given on a 7-point Likert-type scale, from 0 (never) to 6 (always-every day). The reliability and validity of the UWES was analyzed cross-culturally by the authors, and the measure has satisfactory psychometric properties.

The current study also utilized a demographic questionnaire gathering the following data: penitentiary unit type (open, semi-open, closed), education, gender, and age. It also involved questions about shift work (single- vs. multi-shift) and whether the respondent comes from a family of uniformed services members (occupational inheritance).

In order to answer the research questions, statistical analyses were carried out using the IBM SPSS Amos software (24.0 Microsoft Windows, New York, NY, USA). Basic descriptive statistics, Student’s *t* tests for independent samples, the bootstrapping method, path analysis, and multi-group invariance analysis were calculated. The statistical significance threshold was set at *p* < 0.05. Results on a level of 0.05 < *p* < 0.1 were interpreted as a statistical trend.

## 3. Results

### 3.1. Participant Characteristics

The current study involved POs in Poland and in Indonesia. The participatns were informed about the study aims and the option to opt out of participation at any point. The participants gave oral informed consent. The study involved officers of the Prison Service in Poland (*n* = 312, mean age = 34.35 ± 7.35 years) and in Indonesia (*n* = 467, mean age = 34.60 ± 9.91 years). Table 1 presents the specific characteristics of the study group.

### 3.2. Work Engagement and Subjective Well-Being among POs

In the first step, mean work engagement scores on the UWES-9 and subjective well-being assessed via Cantril’s Ladder were calculated. The results are presented in Table 2. The differences between the countries were revealed to be statistically significant. On average, Indonesian POs reported higher subjective well-being and higher work engagement, both in terms of the general score as well as the individual subscale scores.

In order to establish the relationships between the variables, a path analysis was carried out. The analysed model assumed a multivariate normal distribution (C.R. = 5.66) and did not contain missing data. The maximum likelihood (ML) estimation was used for the analyses. The model included the covariation between the factors of coping styles. It was characterized by a good fit to data, χ^2^/df = 4.78; CFI = 0.937; SMRM = 0.064; RMSEA = 0.071. Standardized regression coefficients are shown in Figure 1.

The analysis revealed that shift work (β = 0.04; *p* = 0.164), age (β = −0.06; *p* = 0.078), and occupational inheritance (β = −0.04; *p* = 0.189) were not related to subjective well-being. Only the factor of penitentiary unit type was positively, though weakly, related to subjective well-being (β = 0.11; *p* < 0.001). This means that subjective well-being was higher among POs working in closed prisons compared to semi-open prisons. Regarding coping strategies, statistically significant relationships with subjective well-being were observed for active coping (β = 0.21; *p* < 0.001), avoidant coping (β = −0.37; *p* < 0.001), as well as seeking social support and emotion-focused coping (β = 0.17; *p* < 0.001). The strongest relationship was observed between avoidant coping and subjective well-being—it was negative and moderate, meaning that the higher the frequency of avoidant coping, the lower the subjective well-being. Active coping and seeking social suport and emotion-focused coping were positively and weakly related to subjective well-being, meaning that the higher the frequency of using those coping strategies, the better the subjective well-being.

Penitentiary facility type (β = −0.02 *p* = 0.419), age (β = −0.03; *p* = 0.327), and occupational inheritance (β = −0.03; *p* = 0.389) were not significantly related to work engagement. Only multi-shift work was positively, though weakly, related to work engagement (β = 0.10; *p* < 0.001). Work engagement was higher among POs working in the multi-shift system. From among the analyzed coping factors, only two—active coping (β = 0.18; *p* < 0.001) and social support seeking and emotion-focused coping (β = 0.32; *p* < 0.001), as well as subjective well-being (β = 0.26; *p* < 0.001) were significantly related to work engagement. The higher the level of active coping, social support seeking and emotion-focused coping, and subjective well-being, the higher the work engagement. Avoidant coping (β = −0.05; *p* = 0.104) was not related to work engagement.

Additionally, using the bootstrapping method for 5000 samples, confidence intervals for the indirect effect were calculated in order to examine whether subjective well-being is a moderating variable in the analyzed model. The analysis showed a statistically significant indirect effect of subjective well-being for the relationship between penitentiary unit type (β = 0.03), active coping (β = 0.03), avoidant coping (β = 0.03), and social support seeking and emotion-focused coping (β = 0.03) and work engagement. Subjective well-being was a mediator of all of these variables (see Table 3).

### 3.3. Comparison of the Model between the Countries

In the next step of the analysis, the above model was tested separately in the group of Polish (*n* = 311) and Indonesian (*n* = 467) POs. Multi-group invariance analysis was employed to compare the models between the groups [62]. This method tests the difference between the χ^2^ values. The analysis revealed statistically significant differences between the models for Polish and Indonesian POs, Δχ^2^ = 97.65; *df* = 15; *p* < 0.001; ΔNFI = 0.072. Thus, it can be assumed that the cross-national differences for the examined model are significant. Table 4 shows the comparison of standardized values for the models in each group. Figure 2 shows the standardized regression coefficients in the group of Polish and Figure 3 shows the standardized regression coefficients in the group of Indonesian Pos.

The analysis showed that both groups differed in regard to the relationship between the penitentiary unit type and subjective well-being. Among Polish POs, the relationship between these variables was positive and weak, while among Indonesian POs, these variables were independent from each other. Another differenec concerned the relationship between social support seeking and emotion-focused coping and subjective well-being. For Polish POs, the relationship between these variables was positive, though its statistical significance was at the level of a statistical trend, whereas for Indonesian POs, these variables were not related. A statistically significant difference also emerged for the relationship between subjective well-being and work emgagement—it was positive and moderate for Polish POs, but not statistically significant for Indonesian POs. Similarly, the relationship between multi-shift work and work engagement was positive and weak for Polish POs and not statistically significant for Indonesian POs. The relationship between active coping and work engagement was statistically significant in both groups, though it was stronger in the Polish group. The differences between the models regarding the other variables were not statistically significant.

Additionally, using the bootstrapping method for 5000 samples, confidence intervals for the indirect effect were calculated in order to examine whether subjective well-being is a moderating variable in the analyzed model. The analyses were carried out separately for each group. They showed a statistically significant indirect effect of subjective well-being for the relationship between penitentiary unit type (β = 0.08), active coping (β = 0.07), and avoidant coping (β = −0.10) and work engagement in the group of Polish POs. Subjective well-being was a mediator of these variables. In the group of Indonesian POs, all effects were not statistically significant—subjective well-being was not a mediator (see Table 5).

## 4. Discussion

The current study aimed to analyze the phenomenon of work engagement and its determinant variables, such as subjective well-being, organizational factors, or coping strategies, in a sample of POs. From the point of view of realizing the tasks of the prison service, work engagement and the health of its employees is undeniably important. However, detailed data on the work engagement of prison staff has been unavailable thus far. Additionally, reports on the psychological characteristics of the work conditions of Indonesian POs are scarce. The current study revealed that mean work engagement scores of Polish POs were lower than those of Polish physiotherapists, police offiecrs, or teachers assessed by the same measure [63]. Comparing the detailed results of the Indonesian group, only the scores on the absorbtion subscale were lower for POs than for police officers. Both Polish and Indonesian POs also achieved lower scores on the three UWES-9 subscales compared to psychiatric nurses’ work engagement scores [64].

Although the problems faced by Indonesian penitentiary units are often described in the literature as difficult, the mean subjective well-being scores of Indonesian POs were higher than those of Polish POs. This is a surprising result considering the struggles of the Indonesian prison system with the issues enumerated in the current article’s introduction, as well as, for example, the issue of carrying out the death sentence [65,66]. These issues are absent from the Polish prison system. Few research reports can be found in the literature, which point towards the experience of somatic and psychological stress symptoms among Indonesian POs (heart palpitations, muscle pains, fatigue, or boredom) [67]. Similar research carried out in Poland clearly shows that a significant part of this profession experiences health problems [68].

Among the coping strategies included in the analyses, the strongest relationship emerged between subjective well-being and avoidant coping. According to the division of the COPE factors, this included: behavioral disengagement, humor, denial, substance use, and acceptance. The analyses showed the negative influence of these coping strategies for well-being and job performance. This is in line with the role given to these strategies in the literature. It shows that, aside from facilitating a low level of mental toughness, avoidant coping is also related to the emergence of occupational burnout, among others [69]. Studies on nurses and social workers show that passive behaviors in situations of stress and lacking social support can cause emotional exhaustion with one’s work [70,71]. Avoidant behaviors can take the form of, for example, procrastination or distraction, and thus, are described as maladaptive [72]. Susbstance use is also an evidently unhealthy reaction to stress. Considering the health risks faced by POs, as described in the introduction, effective coping strategies in work situations seem crucial for maintaining mental hygience and good well-being. Difficulties in coping with workplace stress predispose towards poorer health, including pain symptoms and the common cold [73]. Meta-analyses of the effectiveness of psychological interventions for workplace stress unambigiously show that cognitive-behavioral, that is, goal-oriented techniques are the most helpful [74]. Thus, the largest effect in the current study, that is, the relationship between active coping and work engagement, complements the presented literature review. An active approach in the face of difficulties allows to develop solutions and facilitates positive outcomes. As was indicated in the introduction, work engagement involves goal orientation rather than withdrawal.

Subjective well-being was revealed to be moderately related to work engagement. This relationship emerged only in the group of Polish POs. This is in line with prior results, showing that those employees who identify with their organization report better psychological and physical well-being, lower levels of subjectively experienced stress and anxiety, as well as generally better mental health and less frequent pain symptoms [75,76,77]. Work engagement is a negative predictor of health problems. A related line of research reported in the literature concerns the health consequences of workplace bullying [78]. The prison environment is not free of this type of negative phenomena.

Further analyses revealed subjective well-being to be a significant mediator in the relationship between the penitentiary unit type, individual coping strategies, and work engagement. This means that subjective well-being mediates this relationship. In an intergroup comparison, this effect was not statistically significant for the Indonesian POs. Thus far, mental and physical well-being have been included in research models as the explained or independent variable in relation to workplace factors (occupational stressors, engagement, or job performance) [79,80]. However, research shows that well-being can mediate the relationship between workplace stressors and turnover intention [81]. It was also proved that exhaustion, which can be considered in terms of subjective well-being, mediates the relationship between job demands and satisfaction [82]. The above results are an argument in favor of the claim that workplace characteristics (e.g., a given penitentiary unit type and exposure to specific risks) influences employee attitudes (their attitudes towards their job or potential engagement). Additionally, employee mental and physical state plays an important role in this context. It seems obvious that health is simultaneously a value and a resource for optimal functioning in all areas of activity. Good health allows for satisfying numerous needs, including the need for activity, and generates positive mood states.

The current study yielded results which require further development. Differences in subjective well-being between staff of various penitentiary unit types were small, though statistically significant. They showed that POs in closed prisons more often declare high subjective well-being. In both prison systems included in the study, the open character of penitentiary units means a minimal supervision of employees and higher freedom of inmate movement [83]. Simulatenously, studies show that, compared to maximum- and medium-security prisons, POs in minimum-security prisons experience higher levels of job satisfaction and report less health problems: back pain, insomnia, the common cold, anxiety, or boredom [84]. In Polish prisons, multi-shift work is related to more days off work. This might be related to the influence of the possibility of authentic recovery from the impact of stressors, described in the literature [85]. Additionally, night-shift work is less demanding due to a lower level of responsibilities.

## 5. Conclusions

Work engagement among Polish and Indonesian POs differs and is determined by different factors. The current study showed that subjective well-being and specific strategies of coping with stress are significant factors differentiating the attitude towards work. Active coping was revealed to be a significant, positive variable, which is in line with its descriptions in the literature. Surprisingly, despite unfavorable conditions in Indonesian prisons, Indonesian POs achieved higher mean work engagement and subjective well-being results. Moreover, POs working in higher-security units more frequently reported better subjective well-being. The current study provides arguments that work engagement is a complex psychological phenomenon. There exists a justified need to consider both individual determinants (e.g., coping strategies) as well as organizational factors characterizing the POs’ workplaces in future studies. The subject literature provides data on the wide spectrum of benefits stemming from researching this phenomenon.

## 6. Strengths, Limitations, and Future Research

The strengths of the current study include a large sample size from countries with different penitentiary systems. It analyzed the topic of work engagement, which, thus far, has not been researched in detail in the context of POs. Another strength is also the use of measures with good psychometric properties.

However, the study also has several limiations. Due to organizational and legal restrictions, it was carried out in few penitentiary units and custody suites. Increasing the number of facilities included in the study would improve the generalizability of the results. Further considerations about which determinants of work engagement to include are also pertinent. An engaged attitude towards a highly formalized and repressive prison system can be subject to unique determinants [86]. It is likely that temperamental factors determine more optimal functioning in such an environment and engagement despite unfavorable conditions present in such facilities. Continuing the research in this direction is justified by the fact that employee engagement has a proven effect on inmate violence. This points to the importance of further empirical explorations and analyses [87].

Additionally, it has to be highlighted that the current study utilized the simple, graphical measure of Cantril’s Ladder for measuring subjective well-being. It can also be measured with more detailed and objective screening tools (e.g., the GHQ-12, GHQ-28, Quality of Life Inventory, and SF-36 questionnaires) [88]. Both perspectives—the one used in the current study and the one recommended for future studies—are related to the broad conceptualization of health, as defined by the World Health Organizaiton [89]. It would be useful for future studies to employ multiple measures to better explore the subject matter. Future studies can also investigate whether the results of the current study can be replicated in other cultural settings.

## Figures and Tables

**Figure 1 ijerph-17-08206-f001:**
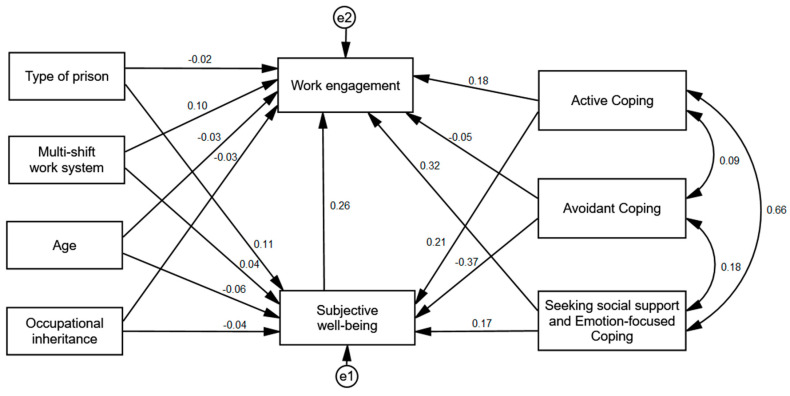
Standardized regression coefficients of the explanatory path model for work engagement.

**Figure 2 ijerph-17-08206-f002:**
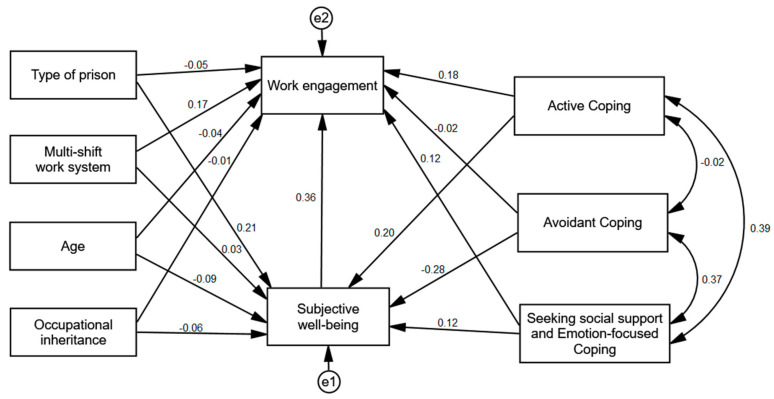
Standardized regression coefficients for the explanatory path model of work engagement in the group of Polish POs.

**Figure 3 ijerph-17-08206-f003:**
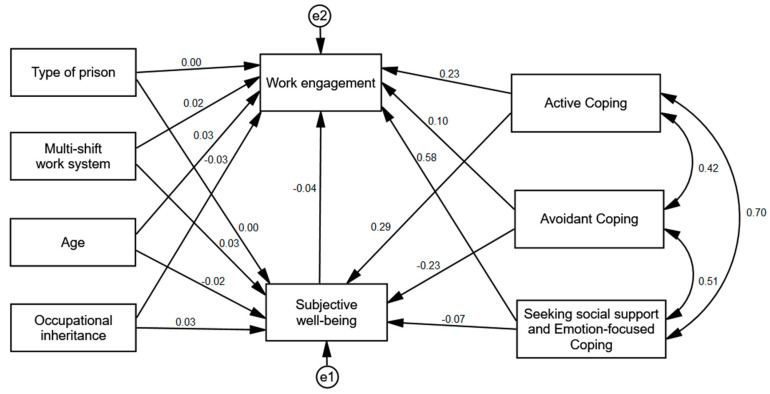
Standardized regression coefficients for the explanatory path model of work engagement in the group of Indonesian POs.

**Table 1 ijerph-17-08206-t001:** Descriptive statistics of the surveyed respondents from Poland and Indonesia.

Variables		Country	
		Poland	Indonesia
	N	312	467
	Age	34.35 ± 7.35	34.60 ± 9.91
	Female	35 (11.4)	103 (21.1)
Education	Secondary education	159 (50.8)	199 (42.6)
	Bachelor’s degree	63 (20.1)	231 (49.5)
	Master’s degree	91 (29.1)	37 (7.9)
Type of penitentiary unit	Open	3 (1.0)	0 (0.0)
	Semi-open	61 (19.7)	69 (14.8)
	Closed	245 (79.3)	398 (85.2)
	Multi-shift system	222 (71.4)	212 (45.5)
	Occupational inheritance	84 (26.9)	198 (42.4)

Note. Percentages are given in brackets.

**Table 2 ijerph-17-08206-t002:** Comparison of subjective well-being and work engagement among Polish and Indonesian prison officers (POs).

Variables	Country	M	SD	T	P
Subjective well-being	Poland	6.96	2.05	−10.58	0.000
	Indonesia	8.57	1.35		
Vigor	Poland	2.81	1.46	−11.18	0.000
	Indonesia	3.64	0.55		
Dedication	Poland	3.10	1.45	−12.67	0.000
	Indonesia	4.02	0.51		
Absorption	Poland	2.69	1.40	−8.97	0.000
	Indonesia	3.33	0.57		
Work engagement	Poland	2.87	1.29	−12.39	0.000
	Indonesia	3.66	0.42		

**Table 3 ijerph-17-08206-t003:** Indirect effects of subjective well-being in the explanatory model of work engagement.

Variables				95% CI for B	
	*β*	*B*	*SE*	*LL*	*UL*
Penitentiary unit type	0.03	0.21	0.08	0.075	0.405
Multi-shift work	0.01	0.07	0.05	−0.025	0.175
Age	−0.01	−0.01	<0.01	−0.010	0.001
Occupational inheritance	−0.01	−0.06	0.05	−0.177	0.028
Active coping	0.05	0.06	0.02	0.032	0.097
Avoidant coping	−0.10	−0.11	0.02	−0.157	−0.068
Social support seeking and emotion-focused coping	0.05	0.06	0.02	0.029	0.113

**Table 4 ijerph-17-08206-t004:** Comparison of standardized regression coefficients for the paths in the model for Polish and for Indonesian POs.

X	Y		Poland			Indonesia		
		*B*	*β*	*P*	*B*	*Β*	*p*	*Z-score*
Penitentiary unit type	Subjective well-being	1.06	0.21	<0.001	0.01	<0.01	0.941	−3.30 ***
Multi-shift work	Subjective well-being	0.13	0.03	0.593	0.07	0.03	0.545	−0.20
Age	Subjective well-being	−0.03	−0.09	0.079	<0.01	−0.02	0.658	1.45
Occupational inheritance	Subjective well-being	−0.27	−0.06	0.262	0.09	0.03	0.483	1.32
Active coping	Subjective well-being	0.20	0.20	<0.001	0.15	0.29	<0.001	−0.67
Avoidant coping	Subjective well-being	−0.21	−0.29	<0.001	−0.15	−0.23	<0.001	1.06
Social support seeking and emotion-focused coping	Subjective well-being	0.14	0.12	0.067	−0.06	−0.07	0.261	−2.15 **
Subjective well-being	Work engagement	0.70	0.37	<0.001	−0.03	−0.04	0.208	−6.67 ***
Penitentiary unit type	Work engagement	−0.51	−0.05	0.307	0.01	<0.01	0.942	1.02
Multi-shift work	Work engagement	1.41	0.17	<0.001	0.06	0.02	0.416	−3.13 ***
Age	Work engagement	−0.02	−0.04	0.458	<0.01	0.03	0.331	0.86
Occupational inheritance	Work engagement	−0.06	−0.01	0.896	−0.07	−0.03	0.279	−0.04
Active coping	Work engagement	0.34	0.18	0.001	0.11	0.23	<0.001	−2.14 **
Avoidant coping	Work engagement	−0.03	−0.02	0.698	0.06	0.10	0.002	1.10
Social support seeking and emotion-focused coping	Work engagement	0.28	0.12	0.049	0.41	0.58	<0.001	0.86

** *p* <0.01; *** *p* < 0.001.

**Table 5 ijerph-17-08206-t005:** Indirect effects of subjective well-being in the explanatory model of work engagement.

Variables				95% CI	
	*β*	*B*	*SE*	*LL*	*UL*
**Polish POs**					
Penitentiary unit type	0.08	0.75	0.24	0.361	1.320
Multi-shift work	0.01	0.09	0.17	−0.229	0.440
Age	−0.03	−0.02	0.01	−0.042	0.001
Occupational inheritance	−0.02	−0.19	0.18	−0.566	0.150
Active coping	0.07	0.14	0.05	0.057	0.249
Avoidant coping	−0.10	−0.15	0.04	−0.228	−0.082
Seeking social support and emotion-focused coping	0.042	0.10	0.06	−0.018	0.235
**Indonesian POs**					
Penitentiary unit type	<0.01	<0.01	0.01	−0.019	0.013
Multi-shift work	<−0.01	<−0.01	0.01	−0.021	0.004
Age	<0.01	<0.01	<0.01	<0.001	0.001
Occupational inheritance	<−0.01	<−0.01	0.01	−0.022	0.004
Active coping	−0.01	−0.01	<0.01	−0.014	0.002
Avoidant coping	0.01	0.01	<0.01	−0.002	0.015
Seeking social support and emotion-focused coping	<0.01	<0.01	<0.01	−0.001	0.010

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
