# Peer review of "Work Engagement among Prison Officers. The Role of Individual and Organizational Factors in the Polish and Indonesian Penitentiary Systems"

_ijerph, 2020, doi:10.3390/ijerph17218206_

Round 1
Reviewer 1 Report
That article is very friutful and interesting, it portreis sagnificant aspect of comparative research. Methodology, for instance diagnostic tools and conception (idea) is deeply analysed. But it is only psychological point of view.
Ofcourse there is point to discuss.
Prison staff form Poland and from Indonesia- but why? The polish and indonesian penitentiary systems are different. Tradition, history and philosophy of penality propobly isn’t similar. So the organizational culture must be different. What was the root of comparison two so different groups?
Characteristic of both penitentiary systems. Only numbers of inmates per 100.000 citizens, the level of occupancy. But it isn’t enought. In my opinion we need theoretical basis. A kind of framework of comaparative research. For instnace - model of prison staff education, recruitment into services, mental and phisical condition (like sport promotion in Czech Republik). What about job perspectives in two places, social respect for the staff? Generally speaking what about organizational culture of penitentiary systems in Poland and Indonesia? It is a part of stressful work, subjective well-being, work environment, strategies.
On the other hand there is description of officers’ groups. It was a great deal of participants. Accordance to my expirience, there are a lot of problmes motivating prison staff for servey. What about the system of motivation. There were no difficulties?
Psychological point of view is very distinct. Ok, but prison Staff is a part of organizational culture.
p. 146- "mean age" among POs in Poland- 4,35? It isn't relevant to table 1.
Author Response
Dear Professor
We are very thankful for a considerate reception of our article and your constructive feedback – science develops only through discussion. Below, we include our responses to the Reviewer’s comments.
- Authors
Q 1
Prison staff form Poland and from Indonesia- but why? The polish and indonesian penitentiary systems are different. Tradition, history and philosophy of penality propobly isn’t similar. So the organizational culture must be different. What was the root of comparison two so different groups?
A 1
We invited colleagues from Indonesia to cooperate for several reasons:
- Our aim was to examine how universal (i.e., independent from the differences in the penitentiary systems) the problems indicated by European and American literature are as the effects of prison officers’ workload. The problems faced by the Indonesian penitentiary system are not characteristic of the conditions in Polish penitentiary facilities. Juxtaposing two significantly different systems was thus deliberate and purposeful on the part of the Authors.
- Indonesian prison officers have not been analyzed in detail with regards to psychological factors, and the scientific literature clearly indicates that the Indonesian penitentiary system is burdened by numerous difficulties.
Q 2
Characteristic of both penitentiary systems. Only numbers of inmates per 100.000 citizens, the level of occupancy. But it isn’t enought. In my opinion we need theoretical basis. A kind of framework of comaparative research. For instnace - model of prison staff education, recruitment into services, mental and phisical condition (like sport promotion in Czech Republik). What about job perspectives in two places, social respect for the staff? Generally speaking what about organizational culture of penitentiary systems in Poland and Indonesia? It is a part of stressful work, subjective well-being, work environment, strategies.
A 2
This is an important remark. A description and comparison of the penitentiary systems is important from the point of view of the reader. Because our article already is 17 pages long (almost 5 thousand words of the main text), we suggest adding a synthetic description of these systems. This information has been added to the article’s body.
Q 3
On the other hand there is description of officers’ groups. It was a great deal of participants. Accordance to my expirience, there are a lot of problmes motivating prison staff for servey. What about the system of motivation. There were no difficulties?
A 3
The Polish authors have cooperated with the Prison Service for several years. This cooperation chiefly involved joint research, training, and publishing. Thus, while conducting research requires time and effort, it is achievable. The Indonesian author is employed by the Prison Service and trains prison officers, which also gives them the opportunity to conduct research.
Q 4
Psychological point of view is very distinct. Ok, but prison Staff is a part of organizational culture.
A 4
This is a valid comment. Additional information has been added to the main text of the article.
Q 5
p. 146- "mean age" among POs in Poland- 4,35? It isn't relevant to table 1.
A 5
All authors have read the final version of the article many times, yet no one has noticed it. This is a valid point; the text has been corrected.
Reviewer 2 Report
Overall, it's a good paper.
It would have been interesting to look at PO's specific sources social support for the PO's like work-family balance.
PO's as an occupation, is one that is a thankless job. PO's keep criminals incarcerated against their will, some of them being willing to harm or even kill PO's. The job of a PO is unlike other jobs where there is some measure of gratitude and recognition from customers. So, sources of social support are few and far between.
Other than occupational inheritance, what draws people to become PO's? Is it the salary for a job that requires minimal education? The benefits? While not the focus of this study, who self-selects into this occupation might be a factor in understanding their level of engagement.
There are a few typos which I noticed throughout, including lines 150 and 331 (participants; limitations, respectively)
The Reference section is not in alphabetical order.
Author Response
Dear Professor
We are very thankful for a considerate reception of our article and your constructive feedback. Below, we include our responses to the reviewer’s comments.
- Authors
Q 1
It would have been interesting to look at PO's specific sources social support for the PO's like work-family balance.
A 1
Social support is crucial in such a difficult line of work. It allows for mitigating workplace stress levels. A balance between work and family life is naturally important. During personal conversations, prison officers have often pointed out their family’s buffering role against the difficulties they experience at work. This is also evidenced by research results. Information on this point has been added to the main text.
Q 2
PO's as an occupation, is one that is a thankless job. PO's keep criminals incarcerated against their will, some of them being willing to harm or even kill PO's. The job of a PO is unlike other jobs where there is some measure of gratitude and recognition from customers. So, sources of social support are few and far between.
A 2
The difficulty of this line of work is exemplified by, among others, the number of physical assaults on penitentiary officers during their work. They rarely gain satisfaction from resocialization work. Contact with aggressive inmates, drastic displays of the prison subculture, a high incidence of repeat offenders, and low social prestige of this position make social support play a particularly important role. Information on this point has been added to the article body.
Q 3
Other than occupational inheritance, what draws people to become PO's? Is it the salary for a job that requires minimal education? The benefits? While not the focus of this study, who self-selects into this occupation might be a factor in understanding their level of engagement.
A 3
Stable employment and wages, additional benefits, the option of earlier retirement, as well as the economic conditions in a given region, for example, the unemployment rate, are additional factors influencing the decision to join the Prison Service. Education – Out of 26890 POs in Poland, 12954 have a Master’s degree, 3701 have a Bachelor’s degree, and 10235 have secondary education. Individuals with secondary education are most often employed in the security sector (guards). No official statistics are available for Indonesia.
Q 4
There are a few typos which I noticed throughout, including lines 150 and 331 (participants; limitations, respectively)
A 4
This is a good point. This has been corrected.
Q 5
The Reference section is not in alphabetical order.
A 5
The positions in the bibliography are arranged in order of citation. For example, line 57 reads: “In turn, problem-solving, seeking instrumental support, or identifying priorities in action have an adaptive character [26].” Position 26 in the bibliography is: “26. Skinner, E. A., Edge, K., Altman, J., & Sherwood, H. Searching for the structure of coping: A review and critique of category systems for classifying ways of coping. Psychol Bull, 2003, 129(2), 216–269; DOI: 10.1037/0033-2909.129.2.216.” This format has been used in recent articles.
Reviewer 3 Report
I thank the authors for their work. This is an interesting study on a subject that has not been widely researched. However, I have some questions and suggestions for the authors.
1- Is there any reason to carry out the study in Poland and Indonesia? Are you trying to compare something?
2- The introduction is a bit reduced but it is well planned. The authors could propose an extension, introducing an extension of the explanation on the variables they have taken into account.
3- In the method section, I suggest that the authors write a paragraph dedicated to the type of research. If it is random, convenience sample, etc ...
4- The authors present an approval by part of the ethics committee. That is very good but it would help to know how the process was. Some part shows up in results, such as how anonymity was maintained. I think that all this process should be described in the method section.
5- The results section is very interesting but, I insist, it must be justified why the sample is directed to Poland and Indonesia and why these two countries are compared. Is it from comparing cultures? By comparing opposing prison systems? This must be introduced at some point in the text to become aware of the reason for this study.
6- Congratulations for the discussion and conclusions made. They are consistent and proven. I greatly appreciate that they have introduced a limitation and prospective section, it is essential for this type of study.
Author Response
Dear Profesor
We are very thankful for a considerate reception of our article and your constructive feedback. Below, we include our responses to the reviewer’s comments.
- Authors
Q 1
1- Is there any reason to carry out the study in Poland and Indonesia? Are you trying to compare something?
A 1
We invited colleagues from Indonesia to cooperate for several reasons:
- Our aim was to examine how universal (i.e., independent from the differences in the penitentiary systems) the problems indicated by European and American literature are as the effects of prison officers’ workload. The problems faced by the Indonesian penitentiary system are not characteristic of the conditions in Polish penitentiary facilities. Juxtaposing two significantly different systems was thus deliberate and purposeful on the part of the Authors.
- Indonesian prison officers have not been analyzed in detail with regards to psychological factors, and the scientific literature clearly indicates that the Indonesian penitentiary system is burdened by numerous difficulties.
Q 2
2- The introduction is a bit reduced but it is well planned. The authors could propose an extension, introducing an extension of the explanation on the variables they have taken into account.
A 2
This is an important remark. Describing the variables is significant from the point of view of the reader. Because our article is already 17 pages long (almost 5 thousand words of the main text), we suggest the addition of a synthetic explication and description of the variables. If the Reviewers and Editors agree, we can additionally expand the text.
Q 3
3- In the method section, I suggest that the authors write a paragraph dedicated to the type of research. If it is random, convenience sample, etc ...
A 3
Thank you for this comment. This has been added.
Q 4
4- The authors present an approval by part of the ethics committee. That is very good but it would help to know how the process was. Some part shows up in results, such as how anonymity was maintained. I think that all this process should be described in the method section.
A 4
The description has been moved to the Method section.
Q 5
5- The results section is very interesting but, I insist, it must be justified why the sample is directed to Poland and Indonesia and why these two countries are compared. Is it from comparing cultures? By comparing opposing prison systems? This must be introduced at some point in the text to become aware of the reason for this study.
A 5
The comparison was made on the basis of the differences between the penitentiary systems. They have been outlined in the article’s Introduction. The Discussion section attempts to analyze which variables included in the study are universal, and which differentiate the groups of participants from individual countries.
Q 6
6- Congratulations for the discussion and conclusions made. They are consistent and proven. I greatly appreciate that they have introduced a limitation and prospective section, it is essential for this type of study.
A 6
Thank you